# Qualitative Factor-Based Comparison of NMR, Targeted and Untargeted GC-MS and LC-MS on the Metabolomic Profiles of Rioja and Priorat Red Wines

**DOI:** 10.3390/foods9101381

**Published:** 2020-09-29

**Authors:** Dimitrios Kioroglou, Albert Mas, Maria C. Portillo

**Affiliations:** Department Bioquimica i Biotecnologia, Universitat Rovira i Virgili, 43007 Tarragona, Spain; dimitrios.kioroglou@urv.cat (D.K.); albert.mas@urv.cat (A.M.)

**Keywords:** NMR, GC-MS, LC-MS, wine ageing, metabolomics

## Abstract

Wine origin and ageing are two factors related to wine quality which in turn is associated to wine metabolome. Currently, new metabolomic techniques and proper statistics procedures allow accurate profiling of wine metabolome. Thus, the main goal was to evaluate different metabolomic methodologies on their ability to provide patterns on the wine metabolome based on selected factors, such as ageing of barrel-aged wine (factor time), prior usage of the barrels (factor barrel-type), and differences between wine ageing in barrels or glass bottles (factor bottled-wine). In the current study, we implement NMR, targeted and untargeted GC-MS and LC-MS metabolomic analytical techniques so as to gain insights into the volatile and nonvolatile wine metabolome composition of red wines from two cellars located in the only two Spanish Qualified Appellations of Origin; DOQ Priorat and DOCa Rioja regions. Overall, 95 differentially significant metabolites were identified facilitating the evaluation of the analytical methodologies performance and finding common trends of those metabolites depending on the considered factor. The results did not favor NMR as an effective technique on the current dataset whereas suggested LC-MS as an adequate technique for revealing differences based on the factor time, targeted GC-MS on the factor barrel-type, and untargeted GC-MS on the factor bottled-wine. Thus, a combination of different metabolomic techniques is necessary for a complete overview of the metabolome changes. These results ease the selection of the correct methodology depending on the specific factor investigated.

## 1. Introduction

Wine is a complex hydroalcoholic solution including hundreds to thousands of different molecules (e.g., sugars, amino acids, organic acids, lipids, phenolics, alkaloids, sterols, lignans, terpenes, fatty acids). These compounds account for the metabolomic profile of the wine which confers and modulates the quality and sensory properties of the final wine [1,2]. Several factors are involved in the wine metabolome and quality, such as the grape variety, the yeast and bacteria performing the alcoholic and malolactic fermentations, the winemaking practices (e.g., SO_2_ addition, fining agents) and ultimately, the ageing process [1,2,3,4].

The wine ageing process is the period that starts at the end of winemaking with its introduction in wooden barrels and continues after bottling until its consumption. Barrel ageing improves wine stability, color, aroma, and flavor. It is well recognized that the main factors related to the quality of barrel aged wines are the wine composition, ageing time, and wood composition along with its toasting level [3]. In addition, after barrel ageing, factors such as storage conditions, SO_2_ addition, and stopper composition may also influence wine chemical composition during bottle aging [4,5,6].

Overall, the chemical composition of ageing wine reflects the history and conditions during ageing and storage [1]. For instance, it is well recognized that compounds such as 5-methylfurfural are formed during the toasting process and later transferred to the wine during the ageing process, whereas other such as 4-ethylphenol and 4-ethylguaiacol have microbial origin [7,8]. Some authors have investigated the effect of selected factors on the metabolomic profile of wines during ageing using mainly targeted metabolomic or single analytical techniques [3,4,9]. The most widely analytical techniques used in wine metabolomics are gas chromatography-mass spectrometry (GC-MS), high-performance liquid chromatography-mass spectrometry (HPLC-MS), and nuclear magnetic resonance (NMR). GC-MS has preferably been used to profile wine volatile metabolites whereas HPLC-MS has been the most widely used for nonvolatiles, NMR spectroscopy gives a reproducible direct identification and quantification of a broad range of analytes without sample pretreatment [10]. Though, NMR is limited as it is unable to detect metabolites that are present in very small concentrations. Nevertheless, it has been generally accepted that untargeted analysis is needed for a more comprehensive and holistic analysis [1,11]. The use of the kinetic correlations in time-dependent processes as wine ages can further contribute to metabolomic monitoring, discovery of new biomarkers, and metabolic network investigations. Frequently, data are analyzed by multivariate statistical methods, but the choice of the proper statistical treatment plays an important role in drawing conclusions. The most frequently implemented methodologies include principal component analysis (PCA) or principal coordinate analysis (PCoA), correlation analysis, ANOVA, *t*-tests, and hierarchical clustering analysis. However, the reported statistical significance alone does not provide enough evidence for the importance of the findings without estimating the magnitude of the effect [12]. Thus, the determination of the practical significance (effect size) of different conditions or treatments is also of outstanding importance so as to discern the most relevant changes in metabolites.

The aim of this study was to combine different metabolomic analytic techniques (NMR, targeted and untargeted GC-MS, and untargeted LC-MS) to find common patterns on the evolutions of the detected metabolites according to certain selected factors. For that, we analyzed red wines aged in oak barrels from two cellars located in the only two Spanish Qualified Appellations of Origin; DOQ Priorat (Porrera, Catalonia) and DOCa Rioja (Logroño, Spain) regions. The factors considered for the comparison included time of wine ageing in the barrels, prior usage of the barrels and, in the case of Rioja wines, differences between wine ageing in oak barrels or glass bottles. These factors are shortly referred to as time, barrel-type, and bottled-wine, respectively, in the study. Moreover, it needs to be underlined that among the aims of the current study is not the direct comparison of the wine metabolome between the two cellars, as other factors such as grape variety and climatic conditions would render such comparison incoherent. Instead, the study focuses on the comparison between the different metabolomic analytical techniques in relation to their ability to reveal informative patterns regarding the wine metabolome.

## 2. Materials and Methods

### 2.1. Samples

French oak mid-toasted barrels were the source of red wine samples. Two of them are located in a winery of the DOQ Priorat (cellar Ferrer Bobet, FB) and the other two in the DOCa Rioja (bodega institucional, Instituto de Ciencias de la Vid y el Vino, ICVV). In each region the two barrels differed in time of usage, with one barrel being new, without any prior usage (BAN), while the other had been used for one year and is referred as old (BAO). Cleaning of the used barrels was done with the standard cellar practices (washing with pressurized hot water and rinsing). The main parameters of the wine before being introduced in the barrels were similar: 13.8% and 14.1% ethanol; pH 3.3 and 3.4; 0.29 and 0.34 g/L acetic acid; 4.4 and 4.3 g/L tartaric acid (total acidity); 80 and 90 ppm total SO_2_; 1.5 and 1.2 g/L residual sugar; 0.88 and 0.94 g/L malic acid at the end of malolactic fermentation for FB and ICVV, respectively. The barrels followed the habitual cellar management and were maintained with the rest of the barrels from the same vintage. In FB, grape variety was Carignan, which is the main and characteristic variety in DOQ Priorat, and the wine samples were collected at the end of malolactic fermentation inoculated with an autochthonous strain of *Oenococcus oeni*, completed inside BAO and BAN and denoted as 0 time-point, at the time-points of 3, 6, and 9-months of barrel ageing from both barrels, and at the 12-month time-point from BAN only, as BAO was accidentally used to refill other barrels due to common practices in the cellar. On the other hand, the grape variety at ICVV winery was Tempranillo, which is the main variety in DOCa Rioja, and the wine samples were collected at the end of spontaneous malolactic fermentation, completed inside the steel tank and denoted as FML or 0 time-point, and after 3, 9, and 12-months of barrel ageing from both barrels. Additionally, the same day that the wine finished the FML at ICVV winery and transferred into BAO and BAN, a sample of 750 mL from each barrel was taken and bottled into a dark glass bottle as the cellar uses for its wine commercialization. These bottle-aged wine samples, from the old (BTO) and new (BTN) barrel, were stored in the same cellar as the barrels and analyzed after 12-months of bottle ageing. At each sampling point, we sampled three bottles of 50 mL of aged wines with a sterilized pipette of 100 mL introduced into the barrel by a top overture and used for stirring the wine in the barrel and sampling. The schematic diagram of the sampling is presented in Figure 1. All collected wine samples were immediately frozen and preserved at −80 °C prior to analysis. Thus, the sample replicates for the metabolomic analysis were as follows: for the factor “barrel-type” that had two groups in FB and two groups in ICVV (old and new barrel), four replicates represented each group; for the factor “time” that has four groups in FB and four groups in ICVV (corresponding to the sampling timepoints), each group represents an array containing two replicates; ICVV has an additional factor (“bottled-wine”) that included two groups (0 and 12 months) with two replicates per group.

### 2.2. H-NMR

The reagents used were phosphate buffer and phosphoric acid both from Sigma-Aldrich and D2O from CortecNet. For NMR analysis, 450 μL of wine samples were mixed with 250 μL of 1.5 M PBS (pH = 3.2) buffer in D2O. Then, the sample was vortexed and the mixture was centrifuged (15,000 rpm for 15 min at 4 °C) and 600 μL of the clear upper phase was placed into a 5 mm o.d. NMR tube (Eretic Signal 0.6616 mM) and analyzed. ^1^H NMR spectra were recorded at 300 K on an Avance III 600 spectrometer (Bruker^®^; Ettlingen, Germany) operating at a proton frequency of 600.20 MHz using a 5 mm PBBO gradient probe. Wine aqueous samples were measured and recorded in procno 11 using a one-dimensional ^1^H pulse. Experiments were carried out using the nuclear Overhauser effect spectroscopy (NOESY) presaturation sequence (RD-90°-t1-90°-tm-90° ACQ) to suppress the residual water peak, and the mixing time was set at 100 ms. Solvent presaturation with irradiation power of 75 Hz was applied during relaxation delay (RD = 5 s) and mixing time, (noesypr1d pulse program in Bruker^®^) to eliminate the residual water moisture of deuterated water. The acquisition time (ACQ) was 3.42 s for a total recycling delay (RD + ACQ) of 8.42 s, the 90° pulse length was calibrated for each sample and varied from 10.12 to 11.68 ms. The spectral width was 10 kHz (20 ppm), and a total of 256 transients were collected into 64 k data points for each ^1^H spectrum. The exponential line broadening applied before Fourier transformation was of 0.5 Hz. The frequency domain spectra were manually phased and baseline-corrected using TopSpin software (version 3.2, Bruker).

After preprocessing and visually assessing the NMR dataset, specific ^1^H regions of compounds were identified in the spectra using a comparison into AMIX 3.9 software and Chenomx 8.4 software and bibliography. Curated identified regions across the spectra were integrated using the same AMIX 3.9 software package and exported to excel spreadsheet in order to give relative concentrations.

### 2.3. GC-MS

The analytical standards 4-ethylphenol 99%, 4-ethylguaiacol analytical standard, 5-Methylfurfural ≥ 98%, 5-(hydroxymethyl) furfural analytical standard, 2-methylisoborneol solution, certified reference material, TraceCERT^®^, 100 μg/mL in methanol, p-Cresol ≥ 99% (SI) and 3,4-dimethylphenol 98% (SI) were purchased at Sigma-Aldrich, ultrapure water was from an in-house Milli-Q system (Millipore) and methanol LC-MS grade, dichloromethane and ammonium sulfate were from Sigma-Aldrich. For volatile compound extraction a liquid–liquid extraction method with dichloromethane was used. Briefly, 2 g of ammonium sulfate were mixed with 3.15 mL of water, 1.35 mL of wine sample, and 10 μL of IS solution at 100 mg/L and vortexed. Then, 250 μL of dichloromethane was added and mixed for 1 h. The organic layer was collected and transferred to a chromatographic vial for their analysis by GC-MS. GC-MS analysis was performed on a GCxGC-TOF Pegasus 4D from Leco Instruments equipped with MPS autosampler from Guerstel. Chromatographic column was a CP-Sil 24 CB (30 m × 0.25 mm id, 0.25 µm film) from Agilent Technologies. The injection volume was 1 µL and it was performed in pulsed splitless mode in a split/splitless injector at 250 °C. He (99.999%) was used as mobile phase at a constant flow of 1.2 mL/min. For the elution of compounds, the following temperature program was used: 50 °C for 2 min, 50–150 °C at 5 °C/min, 150–240 °C at 10 °C/min. The transfer line temperature was 250 °C and ionization was made by electron impact at 70 eV with a source temperature of 250 °C. The MS acquisition was in full scan after a solvent delay of 5 min between 35 and 600 m/z at 20 scan/seg.

Data analysis for both target and untargeted experiments were performed in Chromatof 4.50.8 software from LECO. For untargeted analysis, the chromatograms were deconvoluted by fixing a baseline offset of 1, a peak width of 1 and signal/noise ratio of 100. For targeted analysis, the method was validated by evaluating the limits of detection (LOD), limits of quantification (LOQ), linearity (R2), recovery, accuracy, and repeatability using standard solutions and standard additions to a representative pool of samples. Quantification of target compounds was performed by an internal standard calibration method whose parameters are detailed at Appendix A.

### 2.4. LC-MS

The reagents used were ultrapure water from an in-house Milli-Q system (Millipore) and methanol LC-MS grade, acetic acid LC-MS grade, formic acid LC-MS grade, ammonium acetate, and ammonium formate were from Sigma-Aldrich. For untargeted LC-MS analysis, the wine samples were filtered (0.22 μm, Nylon) and transferred to an amber glass vial and directly analyzed on a UHPLC-qTOF 6550 from Agilent Technologies. Chromatographic column was an Aquity BEH-C18 (100 × 2.1 mm, 1.7 µm) from Waters. For the elution of compounds, two different chromatographic methods were used with a 5 mM ammonium formate (pH = 3.8) for positive ionization mode and 5 mM ammonium acetate (pH = 4.5) for negative ionization mode as aqueous mobile phase and pure methanol as organic mobile phase component in both methods. The elution gradient was the same for both mobile phases, consisting of 0–0%, 1 min; 0–65% 7 min; 65–100% 8 min; 100–100%, 11 min. The injection volume was 1 µL, the flow rate was 0.6 mL/min, and column temperature was 40 °C. The ionization was performed both in positive and negative electrospray in two separate runs and mass spectra was recorded between 100 and 1100 m/z at 3 spec/seg.

Data analysis was performed with Mass Profinder Software from Agilent. This software deconvolutes the chromatograms to find the molecular features present in the samples and align their mass and retention times resulting in a matrix containing the neutral mass of the feature, their retention time, and the area of the chromatographic peak. For tentative identification of phenolic compounds, the neutral mass obtained for the molecular features found both in positive and negative ionization was matched against phenolic database (http://phenol-explorer.eu/) using Mass Hunter software and allowing a maximum deviation of 20 ppm.

### 2.5. Chemical Classes

The assignment of chemical classes to the identified metabolites was based on the food database (fooddb.ca), the yeast metabolome database [13], and the human metabolome database [14].

### 2.6. Statistical Analysis

Statistical analysis was based on the factors barrel-type and time. For FB, the factor barrel-type included the 0, 3, 6, and 9-month barrel-aged wine samples separated in the groups of old and new barrel resulting in four samples per group, whereas the factor time concerned the barrel-aged wine from old and new barrel grouped by the attributes 0, 3, 6, and 9-month time-points leading to two samples per group. Similarly, for ICVV the factor barrel-type concerned the 3, 9, and 12-month barrel-aged wine samples divided into the groups of old and new barrel, and the factor time comprised the four groups of 3, 9, 12-month barrel-aged and 12-month bottle-aged wine samples. Moreover, ICVV included the additional factor bottled-wine which included the 12-month barrel-aged and 12-month bottle-aged wine samples.

For the analytical methods NMR and targeted GC-MS, statistical significance for each metabolite was derived from Student’s *t*-test based on the factor barrel-type and ANOVA on the factor time using the Python module STATSMODELS [15]. The resulting *p*-values were FDR-corrected (*q*-values) and the statistical significance (*q*-value ≤ 0.05) was coupled with practical significance which is defined as a minimum 2-fold change between minimum and maximum value observed among the samples (FCMM ≥ 2).

For the methods LC-MS and GC-MS, differential metabolomic analysis was performed using the R package MetaboDiff [16]. The analytical steps followed by MetaboDiff included the imputation of missing values using k-nearest neighbor imputation, k-means clustering outlier detection, variance stabilizing normalization, and differential analysis based on Student’s *t*-test or ANOVA using the factors barrel-type and time, respectively. The metabolites that were chosen for further analysis were those that presented statistical significance with FDR-corrected *p*-value ≤ 0.05 and practical significance with FCMM ≥ 2.

Principal coordinate analysis (PCoA) for the methods NMR, LC-MS, and GC-MS was based on the Euclidean distance and permutational multivariate analysis of variance (PERMANOVA) on the distance matrix was performed using the Python module SCIKIT-BIO [17] and the factors barrel-type and time. Finally, hierarchical clustering was performed using the Python module SCIPY [18] after calculating the growth rates of the barrel-aged wine samples for the periods 0–3, 3–6, 6–9, and 9–12-month time-points for FB, and 0–3, 3–9, and 9–12-month time-points for ICVV. For comparison between ICVV’s 12th-month bottle and barrel-aged wine samples, the hierarchical clustering was based on their growth rates for the period 0–12-month. The growth rate between two timepoints t1 and t2, represents a coefficient k so that: k * C1 = C2, where C1 is the concentration of a metabolite at timepoint t1 and C2 its concentration at timepoint t2. The growth rates were calculated as k = 1 + ((C2 − C1)/C1), if there was an increase of concentration between t1 and t2 and as k = 1 + ((C1 − C2)/C1), if there was a decrease of concentration between t1 and t2, so the growth rates were always positive values.

## 3. Results

### 3.1. NMR

The 39 identified metabolites with NMR are reported in Appendix A and Figure 2 for FB and ICVV. For both cellars the concentration for the majority of the metabolites was very low ranging between 0 and 90 mmols with exception ethanol that ranged between 1000 and 1600 mmols. After performing differential analysis, none of the metabolites of FB were found to be statistically significant for the factors barrel-type and time, whereas for the cellar ICVV the metabolite formate was found to be statistically significant for the factor barrel-type and methanol for the factor time. However, methanol had FCMM < 2, leaving formate as the only metabolite with both statistical and practical significance (FCMM = 5.6).

After performing PCoA for the cellar FB (Appendix A), the metabolites with the highest loadings across the principal coordinate were saccharopine, 2,3 butanediol, tartaric acid, and histidine, however, without demonstrating practical significance. The only metabolites with practical significance were formate (FCMM = 2, Appendix A) and ethanal (FCMM = 3, Appendix A), with the former having an impact on separating the early maturation samples (≤3 months) from those of late maturation (≥6 months), and the latter showing differences between old and new barrel for the samples of 3, 6, and 9 months. Nevertheless, their effect on the samples clustering was minimal since PERMANOVA reported nonsignificant differences for the factors barrel-type (*p*-value = 0.46) and time (*p*-value = 0.09), and overall the hierarchical clustering based on the growth rates of the NMR metabolites did not reveal any informative clustering structure (Figure 2A, Appendix A).

Similarly to FB, most of the metabolites of the ICVV’s samples with the highest loadings across the principal coordinates (Appendix A) were practically nonsignificant apart from sorbate (FCMM = 2, Appendix A) that showed differences between early (≤3 months) and late (≥9 months) maturation samples. The rest of the metabolites with practical significance were acetoin (FCMM = 2, Appendix A) with similar PCoA loadings as gluconate that showed differences between early (≤3 months) and late (≥9 months) maturation samples, and formate (FCMM = 5.6, Appendix A) and ethanal (FCMM = 3.5, Appendix A) where both showed differences between old and new barrel and had similar PCoA loadings as sorbate. Practical significance was also shown by uracil (FCMM = 2.4, Appendix A) and shikimic acid (FCMM = 2.0, Appendix A) which had similar PCoA loadings as choline, however, their capacity of demonstrating differences between the factors was minimal and mainly concerned the discrepancy between the 9th month samples with the rest. After performing PERMANOVA, statistical significance was found only for the factor time (*p*-value = 0.02) which upon the calculation of the growth factors was attributed to the higher growth rates of formate and ethanal for the period of 0–3 months of the new barrel and the fact that the majority of the metabolites in both barrels had negative growth rates for the period 9–12 months (Figure 2C, Appendix A).

Overall, the metabolites did not present differences between the 12th month samples of bottle and barrel-aged wine with exception the case of formate that demonstrated practical significance between the 12th month BTN and BAN samples, an exception that also influenced the hierarchical clustering of the samples due to its high growth rate for the BAN sample (Figure 2B, Appendix A).

### 3.2. Targeted GC-MS

The results for targeted GC-MS are given in Figure 3 for FB and ICVV. For FB, the metabolites 4-ethylphenol and 4-ethylguaiacol demonstrated similar trends between old and new barrels, with the former having almost identical values between the barrel-types across the time-points and the latter showing a converging tendency of the barrel-types after the 3rd month (Appendix A). After performing differential analysis, the metabolite 5-methylfurfural showed statistical significance for the factor barrel-type (*q*-value = 0.009) and the metabolite 4-ethylguaiacol for the factor time (*q*-value = 0.002). However, only 5-methylfurfural was considered to present practical significance having a median fold-change of 12.4 between the barrel-types.

Regarding ICVV, the metabolite 5-methylfurfural displayed practical significance of 3.9-fold change difference between the 12th month barrel and bottle-aged wine from new barrel without being accompanied by statistical significance for the factor bottled-wine (Appendix A). However, it presented statistical significance for the factor barrel-type (*p*-value = 0.033) and practical significance of median fold-change of 96.4 between the barrel-types (Appendix A). The rest of the metabolites were statistically nonsignificant and ranged in low concentrations (<23 μg/L).

### 3.3. LC-MS

From the 502 metabolites initially identified with LC-MS, only 14 metabolites were found to be statistically and practically significant for FB (Figure 4) with average FCMM of 6.3 and 17 for ICVV (Figure 5) with average FCMM of 6.

After performing PERMANOVA on the distance matrix for the cellar FB, statistically significant differences were found only for the factor time (*p*-value = 0.01). The impact of the metabolites on the samples clustering (Appendix A) could be distinguished in two groups with the metabolites of each group having similar loadings across the first principal coordinate. The first group contained the metabolites whose concentrations had an increasing trend and had similar PCoA loadings as eriodictyol and the second group those metabolites with a decreasing trend and PCoA loadings similar to isorhamnetin-3-o-glucoside (Appendix A). From the comparison of the groups becomes apparent that the calculated FCMM was derived from the differences between the samples of 0 and 9th or 12th month. This is also being depicted in the hierarchical clustering based on the growth factors where in both barrels the first trimester appears to be the most distant to the rest mainly due to the high growth factors of caffeic acid, jaceosidin, naringin, and luteolin 7-glucoside during that period (Figure 2D, Appendix A).

In the same manner, the LC-MS metabolites for the ICVV samples were divided in those with increasing and those with decreasing tendency (Appendix A), with the metabolites of each group having similar PCoA loadings across the first principal coordinate (Appendix A). Once more, PERMANOVA reported statistical significance only for the factor time (*p*-value = 0.04), and in combination with the hierarchical clustering based on the growth factors it seems that these differences concern the early (≤3 months) and late (≥9 months) maturation periods (Figure 2F, Appendix A). Regarding the bottle and barrel-aged 12th month wine samples, although they demonstrated practically nonsignificant differences with exception the metabolite scopoletin with FCMM of 2 between BAO and BTO and 2.5 between BAN and BTN (Appendix A), minor differences in their growth factors created a cumulative effect able to differentiate the two sample groups (Figure 2E, Appendix A). However, these differences could only be observed across the second principal coordinate of the LC-MS PCoA (Appendix A), that accounted only for 9.5% of the overall observed variation in ICVV’s samples, suggesting weak differences.

### 3.4. Untargeted GC-MS

From the 394 metabolites initially detected with GC-MS, 16 metabolites were found to have statistical and practical significance for FB with mean FCMM of 8.2 (Figure 6) and 48 for ICVV with mean FCMM of 16 (Figure 7).

As with the previous methods, PERMANOVA on the distance matrix showed statistical differences only for the factor time in both cellars (*p*-value = 0.006 for FB and *p*-value = 0.01 for ICVV). For FB three groups of metabolites could be observed influencing the sample clustering (Appendix A). The first group included the metabolites with increasing trend having similar PCoA loadings to diethyl succinate (Appendix A), the second group metabolites with decreasing trend and PCoA loadings similar to indole-3-methyl acetate, and the third group metabolites whose calculated FCMM derived from the differences between the 6th month sample against the rest and had similar PCoA loadings to 4,6,8-trimethylon-1-ene (Appendix A). The latter group appears to be the reason for the clustering of the periods 9–12 and 3–6 months after performing hierarchical clustering based on the growth rates (Figure 2G, Appendix A).

For ICVV, two main groups of metabolites seemed to be influencing the sample clustering (Appendix A). As with LC-MS, the first group demonstrated an increasing trend and had similar PCoA loadings to 2-methyltetrahydrothiophen-3-one (Appendix A) and the second group included metabolites with decreasing tendency and similar PCoA loadings to dibutyl phthalate (Appendix A). These two groups could differentiate the early (≤3) from the late (≥9) maturation wine samples, however, due to the occasional nonlinearity of the trends caused mainly by the 9th month samples in cases such as 6-tridecene (Appendix A), palmitic and stearic acids (Appendix A) the periods 0–3 and 9–12 months clustered together during the hierarchical clustering based on growth rates (Figure 2I, Appendix A).

With regard to the 12th month bottle and barrel-aged wine, a number of metabolites displayed practical significance with a mean FCMM of 7.3 between BAO and BTO and mean FCMM of 6.9 between BAN and BTN that could be separated into two groups. The first group included the metabolites 4,6,8-trimethylnon-1-ene, palmitic acid, stearic acid, 1-dodecanol, and 6-tridecene whose values were higher for the bottle-aged wine samples (Appendix A) and had similar PCoA loadings to 4,6,8-trimethylnon-1-ene (Appendix A). The second group included the metabolites methyl 2-methoxy-2-phenylacetate, oxoglutaric acid, 2,10-dimethylundecane, 5-ethoxyoxolan-2-one, 1-tetradecene, and ethyl-3-hydroxybutyrate that displayed higher values for the barrel-aged wine samples (Appendix A) and received similar PCoA loadings to phenol, 2-(2h-benzotriazol-2-yl)-4,6-bis(1,1-dimethylpropyl) (Appendix A). These two groups appear to mainly have influenced the separation of the 12th month bottle and barrel-aged wine samples after performing growth rates hierarchical clustering (Figure 2H, Appendix A). Additionally, the separation of these two groups of the factor bottled-wine occurs towards the second principal coordinate of GC-MS PCoA (Appendix A) that accounts for 26.2% of the total observed variation suggesting better separation capacity of GC-MS than LC-MS for this factor.

## 4. Discussion

In the current study, the metabolomic profile of barrel-aged wines from two cellars was surveyed by combining NMR, targeted GC-MS, and untargeted GC-MS and LC-MS. The statistical significance (*p*-value or *q*-value ≤ 0.05) was accompanied by practical significance (FCMM ≥ 2) for evaluating the differences between the groups of the factors barrel-type, bottled-wine, and time, since statistical significance alone does not provide enough evidence for the importance of the findings [12].

From previous studies we know that NMR is a useful technique to differentiate vintage, geographical origin, climate, and ageing effect on bottle-aged wine quality [9,19,20,21]. Consonni et al. [9] used NMR to analyze different vintages and ageing times of Amarone wines and found an increase of amino acids during ageing. These authors attributed the increase in amino acids to grape protein degradation ascribed to hydrolysis of yeast and bacteria proteins after their autolysis during the ageing process. Recently, Cassino et al. [21] studied wine evolution during bottle aging and found mostly a decrease in organic acids (lactic acid, succinic acid, and tartaric acid) and an increase in esters (ethyl acetate and ethyl lactate). Furthermore, Catechin and epicatechin decreased during aging in all wines while gallic acid increased in almost all red wines [21]. Nevertheless, in the current study NMR had the least effectiveness in providing informative differences between the groups of the studied factors. From all the metabolites detected by NMR, formate was the only metabolite with both statistical and practical significance for the factor barrel-type in wine samples coming only from ICVV. Although, formic acid has been detected in wine before using NMR [22], it has not been linked previously to barrel ageing.

We designed a targeted GC-MS analysis focused on absolute quantification of 4-ethylphenol, 4-ethylguaiacol, 5-methylfurfural, 2-methylisoborneol, and 5-hidroximethylfurfural as previous studies have related those compounds to wine quality [4,23]. For those studies it is known that there is an enrichment in ethylphenols during wine ageing in barrels but at lower concentration when the barrels are new [3]. Additionally, forced ageing conditions during bottle-aged wine storage resulted in a considerable influence on wine quality increasing the production of dioxane and dioxolane isomers, furfural, and 5-methylfurfural. However, in ICVV and FB samples only 4-ethylphenol, 4-ethylguaiacol, and 5-methylfurfural were detected by this methodology. In fact, both cellars showed significant differences for the concentration of 5-methylfurfural between new and old barrels, having a lower concentration in the latter. This could be reasoned from the fact that this compound is connected with the barrel toasting process justifying its high concentration in the new barrel. Additionally, the high discrepancy of the 5-methylfurfural levels between the new barrels in FB and ICVV could be attributed to the intensity of the toasting process, whereas the detection of 5-methylfurfural in BTN indicates that merely a few hours are enough to transfer this compound to the wine in the case of new barrel and that the compound is stable in the bottle-aged wine even after a 12-month period. Finally, in both cellars the compounds 4-ethylphenol and 4-ethylguaiacol were nonsignificant and ranged below their perception thresholds, 620 and 140 µg/L respectively [24]. Altogether, the results from 5-HMF confirms the results from previous studies while the concentration of the ethylphenols did not change much across barrel ageing or between new and old barrels.

Regarding LC-MS, in both cellars the majority of the identified metabolites were flavonoids demonstrating a decreasing tendency with the exception of the metabolites eriodictyol and jaceosidin in FB and 6-methoxyluteolin in ICVV that increased over time. Total flavonoid content has been reported to decrease after a 70-days storage period [25], however, the temporal concentrations of these three metabolites in the ageing wine have not been monitored before despite the fact that their antioxidant properties have been reported [26,27,28]. Moreover, flavonoid compounds in the wine are represented by groups of flavonols, flavan-3-ols, and anthocyanins. Contact between wine and wood results in a continuous decrease in the anthocyanins content [29] that could be explained by oxidation reactions during ageing or from condensation reactions between anthocyanins and certain wood molecules, all of which generate large, insoluble, and precipitable polymers. The second major group of metabolites identified was benzene derivatives that displayed an increasing trend in both cellars, except the metabolites gallic acid and vanillic acid in ICVV which despite their decreasing tendencies they exhibited overall stable levels. Although the lack of time-series studies of benzene derivatives in ageing wine, studies such as [30] have observed an increase of this chemical class during the late Cabernet Sauvignon grape ripening stage. From the chemical class of cinnamic acids, caffeic acid has been detected with increasing trend in both cellars although it reached higher concentrations in FB. Assuming that the intensity of the toasting process in FB was lower than ICVV, based on the levels of 5-methylfurfural, we could explain this discrepancy of caffeic acid concentrations since studies such as [31] have shown that the content of caffeic acid was significantly lower in toasted French oak woods compared to nontoasted. The same study also reports scopoletin and syringaldehyde of having correspondingly similar and inverse relation to the toasting process compared to caffeic acid. Although these two metabolites have been detected only in ICVV’s wine samples, the similar levels of scopoletin to caffeic acid and the higher concentrations of syringaldehyde in relation to caffeic acid could potentially corroborate these findings. Overall, the two groups of metabolites with temporal increasing and decreasing concentrations, showed statistical and practical significance between the early (≤3 months) and late (>3 months) maturation periods, in both cellars, but revealed no differences for the factor barrel-type. Furthermore, although no statistical and practical significance was reported for the factors barrel-type and bottled-wine, for the latter factor subtle differences between the metabolites growth rates created a cumulative effect able to separate the 12-month bottle-aged wines from the 12-month barrel-aged wines.

As far GC-MS is concerned, a higher variety of chemical classes was identified in both cellars, compared to the other methods, with carboxylic acids being one of the chemical classes that included solely metabolites with increasing trends in both cellars. Among these metabolites was methionol, in ICVV samples, whose degradation has been suggested as a good indicator of oxidation in the wine and that its concentration could be maintained depending on the levels of oxygen and the amount of oxygen scavengers [32]. Given that the level of methionol increased during the first trimester and thereafter remained relatively stable, suggests low initial oxidation levels that increased over time. Another chemical class composed of metabolites with increasing concentrations in both cellars was keto acids with oxoglutaric acid being mutually identified and being described as a metabolite that binds free SO_2_ [33]. The rest of the chemical classes included metabolites that presented both increasing and decreasing trends, that could be explained by small fluctuations of temperature and oxygen levels as well as lysing events. Overall, in both cellars GC-MS did not reveal differences based on the factor barrel-type and had better performance than NMR but worse than LC-MS in clustering the samples in a sensible manner based on the growth rates suggesting an underlying noise. Nevertheless, GC-MS was the only method that detected metabolites with practical significance for the factor bottled-wine, that mainly influenced the separation between the 12-month bottle and barrel-aged wines. Regarding this factor, studies such as [34] have shown significant differences in the aroma after comparing wine ageing in oak barrels and glass bottles with a trained tasting panel. However, in the current study, among these metabolites, the ones that have been connected to aromatic characteristics are palmitic acid, stearic acid, and 1-dodecanol [35,36] all of them having higher concentrations in the bottle-aged wines.

## 5. Conclusions

From the four metabolomic analytical techniques implemented in the current study, NMR was the least effective in providing informative insights based on the given dataset, targeted GC-MS was the only technique that presented significant differences based on the factor barrel-type based mainly on 5-methylfurfural changes, whereas LC-MS and GC-MS were the only methods displaying significant differences for the factor time in both regions. From the latter two methods, GC-MS was also the only one with sufficient separating capacity based on the factor bottled-wine. Nevertheless, no significant differences of metabolites were observed for the factor barrel-type, probably because just one year of barrel usage was not enough to drive differentiation of metabolites at both types of barrels. Currently, the lack of a dedicated open source metabolomic database on the organoleptic characteristics of metabolites renders difficult the inference of the changes imposed on the ageing wine based on the identified metabolites from the untargeted analytical methods. However, the identified metabolites in this study appear to aggregate in two groups; one with increasing and the other with decreasing concentrations and differed significantly between the early and late maturation periods, in both cellars. Thus, common patterns on the metabolites according to the selected factors were discovered even if the origin and characteristics of the wines were different.

## Figures and Tables

**Figure 1 foods-09-01381-f001:**
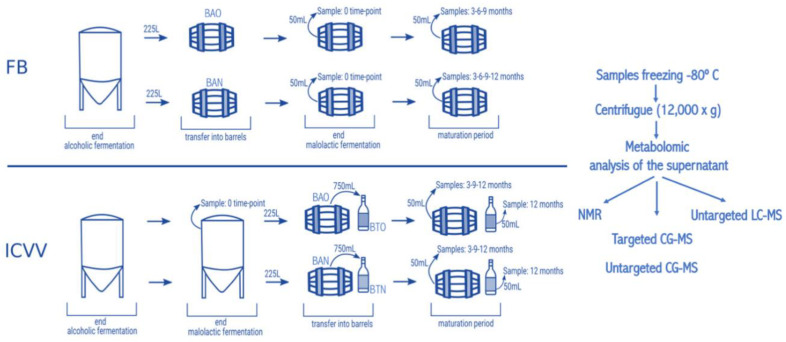
Schematic representation of the experimental setup. The BAN represents new barrels and the BAO the used ones, at the cellar Ferrer Bobet (FB) and cellar of “Instituto de Ciencias de la Vid y el Vino” (ICVV). Barrels of 225 L were sampled at different time-points, expressed in months, taking 50 mL of wine. In FB, the malolactic fermentation was performed inside the steel tank, whereas in ICVV it was performed inside the barrels. In addition, in ICVV the moment the wine was introduced in the barrels, a sample of 750 mL was taken and placed into glass bottles (BTN and BTO). The bottled wines were sampled after 12 months of maturation at cellar conditions.

**Figure 2 foods-09-01381-f002:**
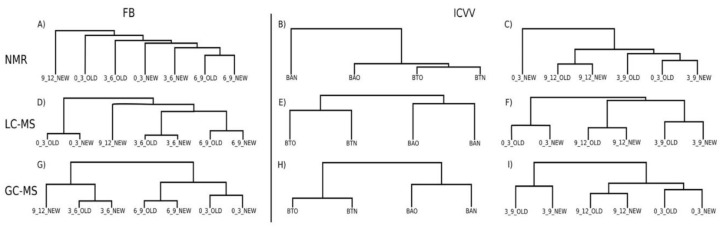
Growth rates hierarchical clustering of samples for region FB (**A**,**D**,**G**) and ICVV (**B**,**C**,**E**,**F**,**H**,**I**) based on different analytical methods. Acronyms BAN and BAO refer to 12th month barrel-aged wine from new and old barrel, respectively, and BTN and BTO to bottle-aged wine from new and old barrel, respectively. Samples containing the labels OLD and NEW refer to barrel-aged wine from old and new barrel, respectively, whereas numbers at the beginning of the labels represent growth rate periods.

**Figure 3 foods-09-01381-f003:**
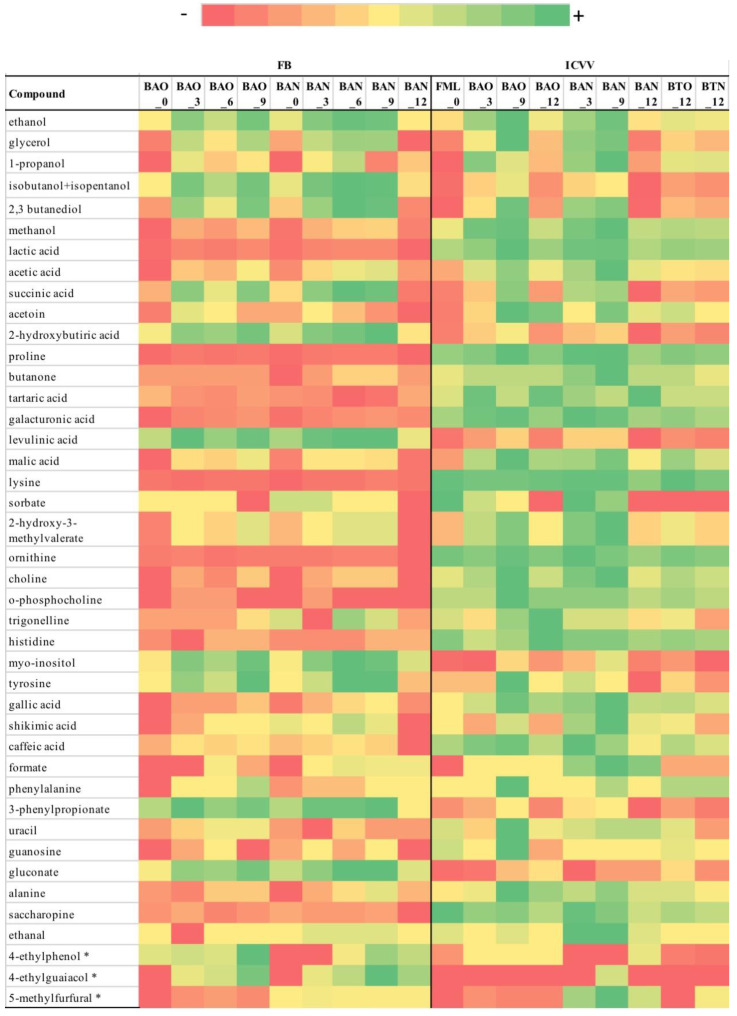
Color coded visualization of Appendix A included in the Appendix A showing the metabolites detected by NMR and targeted GC-MS for cellars FB (**left**) and ICVV (**right**). The color gradient was individualized for the range of detection of each metabolite. Numbers at the end of the sample acronyms represent the sampling month. Targeted GC-MS metabolites are indicated with an asterisk.

**Figure 4 foods-09-01381-f004:**
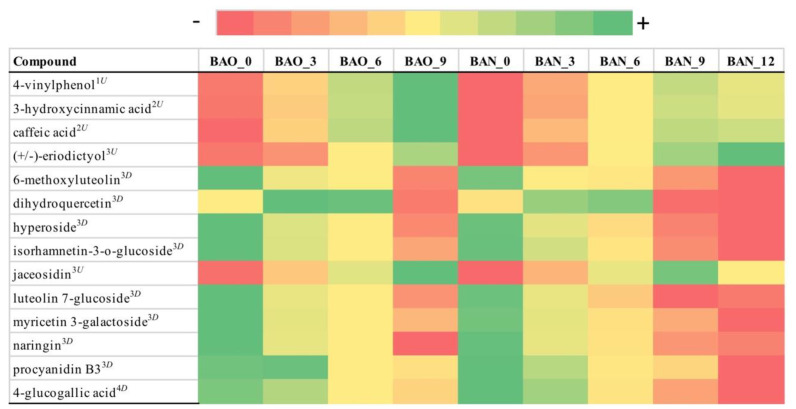
Color coded visualization of Appendix A included in the Appendix A showing the metabolites detected by LC-MS with statistical and practical significance, for cellar FB. Numbers at the end of the sample acronyms represent the sampling month. The color gradient was individualized for each metabolite and the numeric values representing the area of the detected chromatographic peaks after applying variance stabilization are available in Appendix A. The metabolites can be grouped in the following chemical classes: (1) benzene and substituted derivatives, (2) cinnamic acids and derivatives, (3) flavonoids, and (4) organooxygen compounds. Metabolites that exhibited increasing tendency are indicated as superscript (U) and those with decreasing tendency as superscript (D).

**Figure 5 foods-09-01381-f005:**
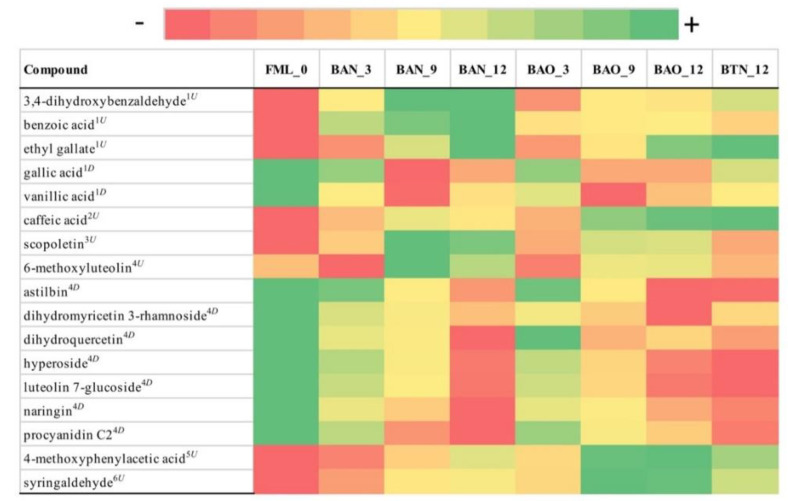
Color coded visualization of Appendix A included in the Appendix A showing the metabolites detected by LC-MS with statistical and practical significance, for cellar ICVV. Numbers at the end of the sample acronyms represent the sampling month. The color gradient was individualized for each metabolite. Additionally, the numeric values represented the area of the detected chromatographic peaks after applying variance stabilization (Appendix A). The metabolites can be grouped in the following chemical classes: (1) benzene and substituted derivatives, (2) cinnamic acids and derivatives, (3) coumarins and derivatives, (4) flavonoids, (5) phenol ethers, and (6) phenols. Metabolites that exhibited increasing tendency are indicated as superscript (U) and those with decreasing tendency as superscript (D).

**Figure 6 foods-09-01381-f006:**
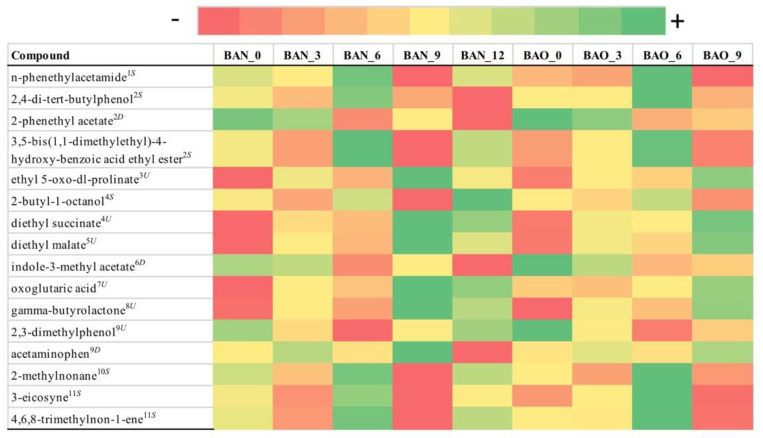
Color coded visualization of Appendix A included in the Appendix A showing the metabolites detected by GC-MS with statistical and practical significance, for cellar FB. Numbers at the end of the sample acronyms represent the sampling month. The color gradient was individualized for each metabolite and the numeric values representing the area of the detected chromatographic peaks after applying variance stabilization are available in Appendix A. The metabolites can be grouped in the following chemical classes: (1) amines, (2) benzene and substituted derivatives, (3) carboxylic acids, (4) fatty acyls, (5) hydroxy acids, (6) indoles, (7) keto acids, (8) lactones, (9) phenols, (10) saturated hydrocarbons, (11) unsaturated hydrocarbons. Metabolites that exhibited increasing tendency are indicated as superscript (U) and those with decreasing tendency as superscript (D), whereas metabolites with relatively stable concentrations but with a peak on the 6th month are denoted as (S).

**Figure 7 foods-09-01381-f007:**
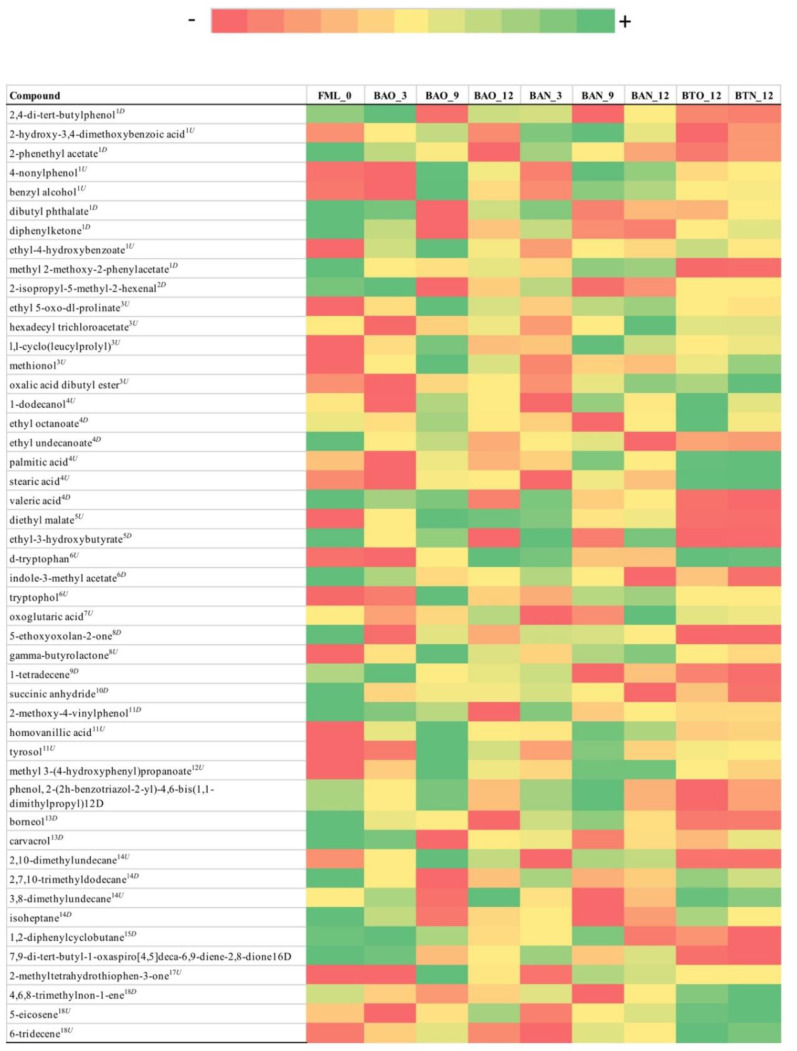
Color coded visualization of Appendix A included in the Appendix A showing the metabolites detected by GC-MS with statistical and practical significance, for cellar ICVV. Numbers at the end of the sample acronyms represent the sampling month. The color gradient was individualized for each metabolite and the numeric values representing the area of the detected chromatographic peaks after applying variance stabilization are available in Appendix A. The metabolites can be grouped in the following chemical classes: (1) benzene and substituted derivatives, (2) carbonyl compounds, (3) carboxylic acids and derivatives, (4) fatty acyls, (5) hydroxy acids and derivatives, (6) indoles and derivatives, (7) keto acids and derivatives, (8) lactones, (9) olefins, (10) oxolanes, (11) phenols, (12) phenylpropanoic acids, (13) phenol lipids, (14) saturated hydrocarbons, (15) stilbenes, (16) tetrahydrofurans, (17) thiolanes, (18) unsaturated hydrocarbons. Metabolites that exhibited increasing tendency are indicated as superscript (U) and those with decreasing tendency as superscript (D), whereas metabolites with relatively stable concentrations but with a peak on the 6th month are denoted as (S).

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
