# Peer review of "Qualitative Factor-Based Comparison of NMR, Targeted and Untargeted GC-MS and LC-MS on the Metabolomic Profiles of Rioja and Priorat Red Wines"

_foods, 2020, doi:10.3390/foods9101381_

Round 1

Reviewer 1 Report

Dear Authors

Thanks for your efforts in this work. The manuscript titled "Qualitative factor-based comparison of NMR, targeted and untargeted GC-MS and LC-MS on the metabolomic profiles of Rioja and Priorat red wines" presents a comparison between three methods of analysis NMR, LC-MS and GC-MS in the metabolome coverage for wine samples from two different sites taking in consideration the time factor, barrel type and bottled wine. The aim of the study is a bit confusing for me specially involving targeted analysis when I think the best way for a holistic metabolome coverage is a high throughput chemical screening which is basically untargeted approach. I don't also see the comparison is sensible because it is anticipated that each of three different methods of analysis could definitely give a unique piece of information taking in consideration that NMR is less sensitive so it is not surprising to have less contribution in terms of number of significantly different metabolites. Please find below some other comments/questions:

1- when you right a decimal so please use dot not comma. So for example 12,3 should be 12.3. Please correct in the whole manuscript

2- Grammar needs to be checked

3- line 36 " It is well recognized" capital (I) so please put dot not comma

4- What is the difference between PCoA (elucidean) and PCA? could you please justify the advantage of using PCoA over PCA?

5- Line 307 correct statical to statistical

6- How did you measure growth rate? what kind of data you have collected for this?

7- How did you identify the metabolites in the tables? have u used LC-MS/MS or authentic compounds?

8- Charts or heatmaps could be more appropriate and informative than tables so please use a graphical presentation for your data to help the reader follow the results and file you tables in SI

9- when you said PCoA loading plot similar to certain metabolite! does this mean you have used reference compound to compare or what did you mean?

10- line 418 "neither of the two metabolites" refers to formic acid and what else?

11- for the targeted approach, should you have clearly mentioned whether the study support the previous hypothesis or not?!

12- for metabolomics setup: how many replicates for each samples? have u used quality control samples? 

Thanks

Best regards

Author Response

(Our answer is shown in italics)

Dear Authors

Thanks for your efforts in this work. The manuscript titled "Qualitative factor-based comparison of NMR, targeted and untargeted GC-MS and LC-MS on the metabolomic profiles of Rioja and Priorat red wines" presents a comparison between three methods of analysis NMR, LC-MS and GC-MS in the metabolome coverage for wine samples from two different sites taking in consideration the time factor, barrel type and bottled wine. The aim of the study is a bit confusing for me specially involving targeted analysis when I think the best way for a holistic metabolome coverage is a high throughput chemical screening which is basically untargeted approach. I don't also see the comparison is sensible because it is anticipated that each of three different methods of analysis could definitely give a unique piece of information taking in consideration that NMR is less sensitive so it is not surprising to have less contribution in terms of number of significantly different metabolites. Please find below some other comments/questions:

Thanks to the reviewer for the appreciation of our work. Regarding the aim of the study, the reviewer is right that we fail to state it clearly in the previous version of the manuscript. We have incorporated now further explanation in lines 10-15 and 84-85 and we think that is clearer now. In short, as commented in the introduction, it is known that several factors affect the wine metabolome and quality during ageing. Thanks to the new metabolomic techniques, accurate profiling of wine metabolome is now possible. Nevertheless, the influence of factors like the time of barrel use, the time of ageing in the barrel and a comparison of the barrel or glass bottle effect on the wine chemistry, have not been evaluated with the new metabolomic techniques following proper statistics procedures. Thus, the questions that we want to answer would be: Are the considered metabolomic techniques useful to detect common patterns in the wine chemistry due to the selected factors even if the wine samples are very different? We assume that different techniques would detect different metabolites and, also it is known that NMR is less sensitive. But what we do not know is if NMR or targeted metabolomic would be enough to detect changes in wine metabolites due to the considered factors.

1.- when you right a decimal so please use dot not comma. So for example 12,3 should be 12.3. Please correct in the whole manuscript.

We have included dots for decimal thorough the article. We are sorry for that mistake.

2- Grammar needs to be checked

Punctuation and grammar have been checked carefully.

3- line 36 " It is well recognized" capital (I) so please put dot not comma

It has been corrected. Thank you.

4- What is the difference between PCoA (elucidean) and PCA? could you please justify the advantage of using PCoA over PCA?

PCA uses a covariance matrix between the metabolites whereas PCoA uses a distance matrix between the metabolites. This distance matrix shows how similar the samples are. Both matrices are calculated based on the concentrations of the metabolites in each sample.  If both methods (PCA and PCoA) use the euclidean distance, then they produce exactly the same plot. In most programs, PCA uses euclidean distance by default. The advantage of PCoA is that the distance matrix can later be the input for PERMANOVA in order to test if there are statistically significant differences based on the calculated distances of the samples for a given factor. PERMANOVA is performed on a distance matrix and not on a covariance matrix.

5- Line 307 correct statical to statistical

It has been corrected. Thank you.

6- How did you measure growth rate? what kind of data you have collected for this?

The data we used were the concentrations in the tables, and the growth rates were calculated for the periods 0-3, 3-6, 6-9 and 9-12-month time-points for FB, 0-3, 3-9 and 9-12-month time-points for ICVV and the period 0-12-month for the bottles.

The growth rate between two timepoints t1 and t2, represents a coefficient k so that:

k * C1 = C2

where C1 is the concentration of a metabolite at timepoint t1 and C2 its concentration at timepoint t2.

The growth rates have been calculated as following:                                                                                      

- If there was an increase of concentration between t1 and t2:

k = 1 + ( (C2 - C1) / C1 )

- If there was a decrease of concentration between t1 and t2:

k = 1 + ( (C1 - C2) / C1 )

We did not want the hierarchical clustering based on the growth rates to be influenced by groups of positive or groups of negative values. Thus, these two equations have been used so that the growth rates have only positive numbers. This information has been now included in material and methods.

7- How did you identify the metabolites in the tables? have u used LC-MS/MS or authentic compounds?

For LC-MS, data analysis was performed with Mass Profinder Software from Agilent. This software deconvolutes the chromatograms to find the molecular features present in the samples and align their mass and retention times resulting in a matrix containing the neutral mass of the feature, their retention time and the area of the chromatographic peak. For tentative identification of phenolic compounds, the neutral mass obtained for the molecular features found both in positive and negative ionization was matched against phenolic database (http://phenol-explorer.eu/ ) using Mass Hunter software and allowing a maximum deviation of 20 ppm.

The information in bold has been included now in the material and methods to complement the missing data.

8- Charts or heatmaps could be more appropriate and informative than tables so please use a graphical presentation for your data to help the reader follow the results and file you tables in SI

The tables have been transformed in color coded visualization and the numeric values have been included as tables in the supplementary information. Frequently, the detected metabolites harbored differences in concentration of several orders of magnitude (up to 5), thus the color gradient has been individualized for each metabolite. Otherwise, the visualization would not added much information as the whole table would be mostly just one color.

9- when you said PCoA loading plot similar to certain metabolite! does this mean you have used reference compound to compare or what did you mean?

In a general sense, the loading refers to the impact that a metabolite has during the clustering of the samples towards a specific direction of the principal components. In the plots we show the metabolites with the highest loadings and an arrow indicating the direction towards which they influence the clustering. By saying “similar loading” between metabolites, we mean that these metabolites have similar clustering impact towards the same direction of the principal component.

10- line 418 "neither of the two metabolites" refers to formic acid and what else?

It refers just to formic acid. Thus, we have change the sentence by “Although, formic acid has been detected in wine before using NMR [22], it has not been linked previously to barrel ageing.”

11- for the targeted approach, should you have clearly mentioned whether the study support the previous hypothesis or not?!

In order to explain clearer whether the targeted approach results support the hypothesis according to previous studies, we have now included additional text indicated here in bold: 

“We designed a targeted GC-MS analysis focused on absolute quantification of 4-ethylphenol (4-EP), 4-ethylguaiacol (4-EG), 5-methylfurfural (5-MF), 2-methylisoborneol (2-MIB) and 5-hidroximethylfurfural (5-HMF) as previous studies have related those compounds to wine quality [4,23]. For those studies it is known that there is an enrichment in ethylphenols during wine ageing in barrels (Chatonnet et al. 1992) but at lower concentration when the barrels are new (Garde-Cerdán et al., 2006). Additionally, forced ageing conditions during bottle-aged wine storage resulted in a considerable influence on wine quality increasing the production of dioxane and dioxolane isomers, furfural and 5-HMF. However, in ICVV and FB samples only 4-EP, 4-EG and 5-MF were detected by this methodology. In fact, both cellars showed significant differences for the concentration of 5-MF between new and old barrel, having lower concentration in the latter. This could be reasoned from the fact that this compound is connected with the barrel’s toasting process justifying its high concentration in the new barrel. Additionally, the high discrepancy of the 5-MF levels between the new barrels in FB and ICVV could be attributed to the intensity of the toasting process, whereas the detection of 5-MF in BTN indicates that merely a few hours are enough to transfer this compound to the wine in the case of new barrel and that the compound is stable in the bottle-aged wine even after a 12-month period. Finally, in both cellars the compounds 4-EP and 4-EG were non-significant and ranged below their perception thresholds, 620 µg/L and 140 µg/L respectively [24]. Altogether, the results from 5-HMF confirms the results from previous studies while the concentration of the ethylphenols did not change much across barrel ageing or between new and old barrels.”

12- for metabolomics setup: how many replicates for each samples? have u used quality control samples?”.

The samples’ replicates for the metabolomic analysis were: for the factor “barrel-type” that had 2 groups in FB and 2 groups in ICVV (old and new barrel), 4 replicates represented each group; for the factor “time” that has 4 group in FB and 4 groups in ICVV (corresponding to the sampling timepoints), each group represents an array containing 2 replicates; ICVV has an additional factor (“bottled-wine”) that included 2 groups (0 and 12 months) with 2 replicates per group.

In order to make it clearer in the article, we have now included this information in the material and methods section.

Thanks

Best regards

Reviewer 2 Report

I propose to do just minor corrections:

1.Potential application of developed method should be mentioned in Abstract.

2. Innovative potential of the results obtained should be explained in detail (CONCLUSIONS).

3. I suggest that a diagram (scheme) presenting the whole analytical process  in the study should be added.  It would help understand the details of the analytical protocol better, and allow the written description of the procedure to be shortened.

Author Response

(Our answers are shown in italics)

I propose to do just minor corrections:

Thanks to the reviewer for the appreciation of our work and the effort of the reviewing.

1.Potential application of developed method should be mentioned in Abstract.

We have now included a potential application of our results at lines 23-25: “Thus, a combination of different metabolomic techniques is necessary for a complete overview of the metabolome changes. These results ease the selection of the correct methodology depending on the specific factor investigated.” Thank you for the suggestion.

  1. Innovative potential of the results obtained should be explained in detail (CONCLUSIONS).

We have now modified the conclusions section both in the abstract and at the end of the manuscript in order to include the innovative potential of the obtained results. Thank you for the suggestion.

  1. I suggest that a diagram (scheme) presenting the whole analytical process in the study should be added. It would help understand the details of the analytical protocol better, and allow the written description of the procedure to be shortened.

A diagram presenting the whole analytical process in our study has been added at the manuscript as Figure 1. Besides, further information on the replicates and methodology has been included in order to ease the understanding of the analytical protocol.

Reviewer 3 Report

The manuscript entitled “Qualitative factor-based comparison of NMR, targeted and untargeted GC-MS and LC-MS on the metabolomic profiles of Rioja and Priorat red wines” described that the change in the metabolite profiles of wines manufactured under different fermentation and ageing conditions using NMR, GC/MS, and LC/MS. The subject matter falls within the scope of the journal “Foods” and there is some interesting data, but some description and data in the manuscript are unclear.

  1. There were so many factors to compare wine quality: 2 wineries (2 countries), 2 different malolactic fermentation places, old and new barrels, bottle and barrel, and 3 different ageing times (3, 6, and 9 months). These many factors prevented a detailed comparison of the quality of wines and description of result, discussion, and conclusions in the manuscript was also very limited. Authors should find out the main factors that affected to wine quality and then rearrange all data with the selected main factors.

  1. There was no information in grape, barrels (did the two wineries use the same barrel?), and fermentation condition including temperature.

  1. How many samples each group were used in this research? The PCA data showed just one sample each group. In general, the number of sample is very important in metabolomic study to get statistically acceptable data.

  1. How did authors identify metabolites analyzed by GC/MS? Did you compare RI values of individual compounds to the values that get using alkanes? Authors should provide detail information about identified compounds, such as MS fragments, MS error, and NMR data.

  1. Tables showed the relative abundance of identified metabolites from each sample group, but it is too difficult to read and compare the values. Please re-visualize these for easy comparison.

  1. The aim of the research is not clear. It is unclear whether the aim of this study is to find the main factors for monitoring wine quality using metabolomic tools or to compare the tools in wine research. Also, what is the comparison of kinetic evolutions of the wine metabolites (page 2, line 65-66)?

Author Response

(Our answers are shown in italics)

The manuscript entitled “Qualitative factor-based comparison of NMR, targeted and untargeted GC-MS and LC-MS on the metabolomic profiles of Rioja and Priorat red wines” described that the change in the metabolite profiles of wines manufactured under different fermentation and ageing conditions using NMR, GC/MS, and LC/MS. The subject matter falls within the scope of the journal “Foods” and there is some interesting data, but some description and data in the manuscript are unclear.

1. There were so many factors to compare wine quality: 2 wineries (2 countries), 2 different malolactic fermentation places, old and new barrels, bottle and barrel, and 3 different ageing times (3, 6, and 9 months). These many factors prevented a detailed comparison of the quality of wines and description of result, discussion, and conclusions in the manuscript was also very limited. Authors should find out the main factors that affected to wine quality and then rearrange all data with the selected main factors.

As explained in the introduction of the article, “among the aims of the current study is not the direct comparison of the wine metabolome between the two cellars, as other factors such as grape variety and climatic conditions would render such comparison incoherent. Instead, the study focuses on the comparison between the different metabolomic analytical techniques in relation to their ability to reveal informative patterns regarding the wine metabolome.” To do so, we choose three different factors that have not been considered before using these metabolomic techniques.

2. There was no information in grape, barrels (did the two wineries use the same barrel?), and fermentation condition including temperature.

Additional information on the parameters of the wines and barrels have now been included in materials and methods to satisfy reviewer’s suggestion. Nevertheless, as explained before, the aim of the study was not the comparison of the wine metabolome between the two cellars that, in fact, were significantly different as expected from previous studies. We search common patters on the wine metabolome of different wines from different regions and under different conditions in response to three common considered factors, using new metabolomic analysis.

3. How many samples each group were used in this research? The PCA data showed just one sample each group. In general, the number of sample is very important in metabolomic study to get statistically acceptable data.

The samples replicates for the metabolomic analysis were: for the factor “barrel-type” that had 2 groups in FB and 2 groups in ICVV (old and new barrel), 4 replicates represented each group; for the factor “time” that has 4 group in FB and 4 groups in ICVV (corresponding to the sampling timepoints), each group represents an array containing 2 replicates; ICVV has an additional factor (“bottled-wine”) that included 2 groups (0 and 12 months) with 2 replicates per group.

In order to make it clearer in the article, we have now included this information in the material and methods section.

4. How did authors identify metabolites analyzed by GC/MS? Did you compare RI values of individual compounds to the values that get using alkanes? Authors should provide detail information about identified compounds, such as MS fragments, MS error, and NMR data.

As explained in the material and methods section, data analysis both for target and untargeted experiments GC/MS were performed in Chromatof 4.50.8 software from LECO. For untargeted analysis, the chromatograms were deconvoluted by fixing a baseline offset of 1, a peak width of 1 and signal/noise ratio of 100 and peak areas were reported. For tentative identification of volatile compounds, the EI spectra of each molecular feature deconvoluted were matched against NIST database. Quantification of target compounds was performed by an internal standard calibration method whose parameters are detailed at supplementary Table 1.

In the case of NMR, after pre-processing and visual checking of NMR dataset, specific 1H regions of compounds were identified in the spectra using a comparison into AMIX 3.9 software and Chenomx 8.4 software and bibliography. Curated identified regions across the spectra were integrated using the same AMIX 3.9 software package and exported to excel spreadsheet in order to give relative concentrations.

For LC-MS, data analysis was performed with Mass Profinder Software from Agilent. This software deconvolutes the chromatograms to find the molecular features present in the samples and align their mass and retention times resulting in a matrix containing the neutral mass of the feature, their retention time and the area of the chromatographic peak. For tentative identification of phenolic compounds, the neutral mass obtained for the molecular features found both in positive and negative ionization was matched against phenolic database (http://phenol-explorer.eu/ ) using Mass Hunter software and allowing a maximum deviation of 20 ppm.

The information in bold has been included now in the material and methods to complement the missing information.

5. Tables showed the relative abundance of identified metabolites from each sample group, but it is too difficult to read and compare the values. Please re-visualize these for easy comparison.

The tables have been transformed in color coded visualization and the numeric values have been included as tables in the supplementary information. Frequently, the detected metabolites harbored differences in concentration of several orders of magnitude (up to 5), thus the color gradient has been individualized for each metabolite. Otherwise, the visualization would not added much information as the whole table would be mostly just one color.

Thank you for the suggestion. We find that the manuscript has improved with that.

6. The aim of the research is not clear. It is unclear whether the aim of this study is to find the main factors for monitoring wine quality using metabolomic tools or to compare the tools in wine research. Also, what is the comparison of kinetic evolutions of the wine metabolites (page 2, line 65-66)?

The reviewer is right. We have rewritten the abstract and also the aim in the introduction in order to make the aim of the study clearer. Abstract: “the main goal was to evaluate different metabolomic methodologies on their ability to provide patterns on the wine metabolome based on selected factors, such as ageing of barrel-aged wine (factor time), prior usage of the barrels (factor barrel-type) and differences between wine ageing in barrels or glass bottles (factor bottled-wine).”. Introduction: “The aim of this study was to combine different metabolomic analytic techniques (NMR, targeted and untargeted GC-MS and untargeted LC-MS) to find common patterns on the evolutions of the detected metabolites according to certain selected factors”.

We have eliminated the “comparison of kinetic evolutions of the wine metabolites” by “common patterns on the evolutions of the detected metabolites”.

Round 2

Reviewer 3 Report

The aim of the revised manuscript is clear and the data are improved. However, so many factors were used to compare analytical tools or wine quality. Authors should find out the main factors that affected to wine quality and then compared analytical tools with the selected main factors.